# Real-World Effectiveness of Racotumomab as Maintenance Therapy in Advanced Non-Small Cell Lung Cancer Patients

**DOI:** 10.3390/vaccines13101035

**Published:** 2025-10-07

**Authors:** Sailyn Alfonso Alemán, Haslen Cáceres Lavernia, Kirenia Camacho Sosa, Soraida C. Acosta Brooks, Orestes Santos Morales, Carmen E. Viada González, Meylán Cepeda Portales, Mayelín Troche Concepción, Loipa Medel Pérez, Leticia Cabrera Benítez, Milagros C. Domecq Salmón, Daymys Estévez Iglesias, Mayra Ramos Suzarte, Tania Crombet Ramos

**Affiliations:** 1Celestino Hernández Robau, Medical Oncology, Santa Clara 50100, Cuba; 2Hermanos Ameijeiras Hospital, Medical Oncology, Havana 10348, Cuba; 3Faustino Pérez Hospital, Medical Oncology, Matanzas 40100, Cuba; 4Saturnino Lora Hospital, Medical Oncology, Santiago de Cuba 90100, Cuba; 5Center of Molecular Immunology, Clinical Direction, Ave 216, Esq. 15, Atabey, Playa, Havana P.O. Box 16040, Cuba

**Keywords:** anti-idiotypic vaccine, racotumomab, NSCLC, cancer vaccine, real world data

## Abstract

Background: Advanced non-small cell lung cancer (NSCLC) has limited curative options and poor survival. Racotumomab, an anti-idiotype monoclonal antibody vaccine targeting tumor gangliosides, has shown efficacy in clinical trials. This study evaluated its real-world effectiveness as maintenance therapy following first-line chemotherapy. Materials and Methods: A multi-center observational study was conducted on 162 patients with advanced NSCLC who received racotumomab from 2012 to 2024. Effectiveness was evaluated in the intention-to-treat (ITT) cohort. Overall survival (OS) was estimated, with subgroup analyses conducted according to clinical and demographic factors. Results: The median OS was 14.9 months (95% CI: 11.7–18.1), and the 5-year survival rate reached 20%. Patients diagnosed with stage III disease, those with better Eastern Cooperative Oncology Group (ECOG) performance status, and individuals younger than 65 years experienced significantly longer survival. Racotumomab demonstrated a favorable hazard ratio compared to historical controls (HR 0.44 vs. supportive care; HR 0.55 vs. docetaxel). Conclusions: In the era of immune checkpoint inhibitors, these real-world results indicate a promising role for racotumomab in the maintenance setting for advanced NSCLC. These findings provide a strong rationale for further investigation of racotumomab in the context of modern immunotherapy, particularly in combination trials with other immunomodulatory antibodies, along with the validation of clinical and biologic predictive biomarkers.

## 1. Introduction

Non-small cell lung cancer (NSCLC) represents a significant global health burden, accounting for approximately 85% of all lung cancer cases and standing as a leading cause of cancer-related mortality worldwide [1]. Its incidence varies geographically and is closely linked to tobacco use [2,3]. Histologically, NSCLC is predominantly composed of adenocarcinoma, which has become the most common subtype, followed by squamous cell carcinoma and large cell carcinoma. In recent decades, epidemiological trends have shown a decline in incidence among men in many high-income countries, while rates among women have risen, partly due to changing smoking patterns and improved diagnostic capabilities [2,4]. The prognosis of NSCLC remains highly dependent on the stage at diagnosis. Early-stage disease is associated with favorable outcomes, with 5-year survival rates exceeding 90% for some subgroups, especially those with ground-glass opacity components [5]. However, a significant proportion of patients are diagnosed at advanced stages, where survival rates drop considerably. The median overall survival for stage IV NSCLC has improved with the advent of targeted therapies and immunotherapy, yet it remains poor, with real-world studies reporting median survival of around 8–10 months in the immunotherapy era [6,7].

The management of advanced disease involves a range of systemic and local modalities, including chemotherapy, radiation, targeted agents, and immunotherapy. For individuals whose tumors harbor specific, sensitizing molecular alterations, the standard first-line approach involves drugs that target these oncogenic drivers, such as Epidermal Growth Factor Receptor (EGFR), anaplastic lymphoma kinase (ALK), the ROS proto-oncogene tyrosine-protein kinase (ROS-1), and Kirsten rat sarcoma (KRAS) [8,9]. Conversely, for patients lacking these actionable mutations, first-line treatment for metastatic cancer is founded on immune checkpoint inhibitors, which may be administered alongside platinum-based chemotherapy or as a single agent [8,9,10].

Cancer vaccines represent an emerging immunotherapeutic strategy for advanced NSCLC, designed to stimulate the patient’s own immune system to recognize and attack tumor cells by targeting specific tumor-associated antigens or neoantigens. Various platforms are under investigation, including personalized neoantigen vaccine such as NEO-PV-01 and UCPVax, which are tailored to individual tumor mutation profiles and have shown promising immunogenicity and safety in early-phase trials, often in combination with chemotherapy or immune checkpoint inhibitors [11,12]. Other approaches include peptide-based vaccines like TG4010 (targeting MUC1) and CIMAvax-EGF, which have demonstrated potential in improving progression-free survival and overall survival in selected patient subgroups [13,14,15].

Gangliosides comprise a large family of glycolipids defined by the presence of neuraminic acid and a location in the outer leaflet of the plasma membrane [16]. These molecules are involved in a diverse range of biological functions, from cell–cell communication and adhesion to regulating the cell cycle and differentiation [17]. They are also recognized for their considerable contribution to tumor progression and the development of metastases [18].

Racotumomab (vaxira and formerly 1E10) is a murine monoclonal IgG1 antibody designed to be used as an anti-idiotypic vaccine against membrane glycoconjugates expressed on solid tumors [16,17,18]. The vaccine is based on Jerne’s idiotypic network theory. It is an anti-idiotype monoclonal antibody (Ab2) that acts as a surrogate for the original tumor antigen (NeuGc-containing gangliosides and sulfated glycolipids) [16,17]. Immunization with this Ab2 induces an Ab3 response (Ab1-like response) that targets the original carbohydrate tumor antigen [19].

Immunization with racotumomab in an alum adjuvant has been shown to induce an anti-metastatic effect in animal models, associated with increased T-cell infiltration into metastases, reduced neovascularization, and enhanced apoptosis of tumor cells in lung nodules [18,20]. Racotumomab has been investigated in several malignancies, including melanoma, breast, colorectal, and small and non-small cell lung cancer [18,20].

A Phase II/III trial investigated the racotumomab vaccine as switch maintenance in 176 advanced NSCLC patients [21]. The vaccinated group demonstrated significantly improved survival, with a median overall survival of 8.23 months versus 6.80 months for placebo (HR 0.63; *p* = 0.004). One- and two-year survival rates were also higher with racotumomab. The vaccine was very well tolerated, with most adverse events being mild or moderate. Based on these results, the National Regulatory Agency approved racotumomab in 2013 [22]. Subsequently, a Phase III randomized controlled trial (RANIDO; IIC-RD 147) compared racotumomab to docetaxel as switch maintenance therapy in advanced NSCLC [14,15]. In this non-inferiority study, the median overall survival was 9.8 months (90% CI 8.8–13.7) and 1-year survival rate was 43.5% for racotumomab, compared to 8.6 months (90% CI 5.9–11.3) and 31.0% for docetaxel, concluding that racotumomab was non-inferior to docetaxel. Once more, the anti-idiotypic vaccine was safe. Common adverse reactions comprised local toxicity at the injection site, myalgia, fevering, and arthralgia. No serious related adverse events (SAEs) appeared in the racotumomab arm, while 4.4% of the docetaxel-treated individuals had related SAEs [22,23].

Real-world data (RWD) refers to information collected through non-interventional methodologies that reflect outcomes in routine clinical practice [24]. RWD complements clinical trials by elucidating treatment performance in broader, more diverse populations typically under-represented in controlled studies. These data are crucial for identifying gaps in care and to evaluate health system performance [25]. The use of real-world evidence (RWE) is increasingly favored by international regulatory agencies to understand drug benefits under routine practice conditions [25]. This manuscript describes a study on the use of racotumomab in routine medical practice to assess its effectiveness.

## 2. Materials and Methods

This was a real-world, retrospective, observational study designed to evaluate the effectiveness of racotumomab as maintenance therapy following first-line chemotherapy in patients with advanced NSCLC. De-identified data were collected from patients treated with racotumomab between 2012 and 2024, at 4 medical oncology services from 4 different provinces. Since racotumomab was a registered drug, patients just signed the institutional informed consent for an approved medical treatment. Confidentiality was granted for all individuals included in the study. The study was approved by the scientific committees of the Center of Molecular Immunology and the participating hospitals.

Eligible patients had a cytologically or histologically confirmed diagnosis of NSCLC and must have completed first-line therapy. Subgroup analyses were performed based on age, sex, skin color, disease stage at diagnosis, histological type, and ECOG performance status [26], to identify potential subpopulations getting the greatest benefit from racotumomab. Data were integrated into a single database managed by the Statistics and Data Management Department of the clinical research direction at the Center for Molecular Immunology (Havana, Cuba).

Treatment was administered as prescribed by the treating physician. The recommended regimen consisted of a 5-dose induction phase (1 mg each administered intradermally every 14 days), followed by re-immunizations every 28 days. Treatment continued until disease progression, unacceptable toxicity, or a clinical decision to halt therapy. Patients could receive subsequent therapy upon progression, at the physician discretion.

Primary outcome was overall survival (OS), defined as the time from vaccination initiation to death or last follow-up contact.

### Statistical Analysis

Patient and tumor characteristics were summarized using descriptive statistics. Effectiveness, analyzed in the intention-to-treat population, was assessed by median overall survival with 95% confidence intervals from Kaplan–Meier curves. Survival distributions were compared using the log-rank test (α = 0.05).

A multivariate Cox proportional hazards regression model was used to assess the independent impact of various clinical and demographic factors on patient survival time. The analysis was conducted by entering all variables, including age (analyzed both as a continuous measure and a binary categorical variable), sex, skin color, smoking habit, ECOG performance status, histologic type, and disease stage simultaneously into the model. Furthermore, a classification and regression tree (CART) analysis was employed to identify subgroups of patients with better survival outcomes based on all baseline variables. This non-parametric method uses a process of recursive binary partitioning to hierarchically split the study population into increasingly homogeneous subgroups. The algorithm selects the cutoff that creates the most distinct groups based on the survival outcome [27]. The CART model employed a growing method with overall survival (continuous) as the dependent variable. Independent variables included age (continuous), sex, skin color, smoking habit, ECOG performance status, histology, and stage, all treated as categorical variables. The maximum tree depth was set to 5 nodes.

To assess the impact of racotumomab in the real-world setting across different subgroups, the study database was integrated with data from the control arms of prior controlled racotumomab trials. The historical data used for contextual comparison were derived from two pivotal, controlled, clinical trials of racotumomab: Phase II/III trial (ID: RPCEC00000009; https://rpcec.sld.cu/trials/RPCEC00000009-En, accessed on 1 September 2025) [21] and the Phase III study (ID: RPCEC000000179; https://rpcec.sld.cu/trials/RPCEC00000179-En, accessed on 1 September 2025) [23]. Data extraction was accurately conducted, with overall survival, patient demographics, and baseline characteristics obtained directly from the original databases. The process was performed by the same statistician involved in the original studies, ensuring consistent and direct oversight. Forest plots were generated to compare outcomes from the real-world cohort with control or docetaxel groups from previous randomized clinical trials [21,23]. IBM SPSS Statistics Version 25 was used for the statistical analysis.

Since the safety of racotumomab was thoroughly evaluated in previous large clinical trials, involving more than 1500 NSCLC patients, the collection of data on adverse events was not an objective of this observational, retrospective study. After the marketing approval, given racotumomab’s favorable safety profile, a passive pharmacovigilance system was implemented, relying on spontaneous reports of serious adverse events.

## 3. Results

Data from 162 NSCLC patients receiving racotumomab from 2012 to 2024, at four clinical sites, were included in this retrospective, real-world study. Figure 1 displays the consort diagram of the study. The median follow-up time of the patients was 112 months. The median age was 65 years, the majority were male (106, 65.4%), and 113 individuals (69.8%) had white skin color (Table 1).

All had advanced disease at diagnosis: 88 (54.3%) had stage IV while 74 (45.7%) had unresectable stage III cancer. Histological types were adenocarcinoma (53 patients, 32.7%), squamous cell carcinoma (51 patients, 31.5%), and large cell carcinoma (45 patients, 27.8%); 13 patients (8.0%) had NSCLC not otherwise specified (NOS). Nearly all patients (95.0%) had a good performance status (ECOG PS = 0: 47 patients, 29.2%; ECOG PS = 1: 106 patients 65.8%) (Table 1).

A hundred and fifty (150) patients (92.6%) completed the 5-dose induction regimen. The median number of doses was 13, corresponding to approximately 10 months vaccination. The main cause of treatment discontinuation was disease progression. There was no treatment interruption due to toxicity. The mean and median overall survival were 32.4 months (95% CI 26.2–38.6) and 14.9 months (95% CI 11.7–18.1), respectively. The 5-year survival rate was 20%.

Survival varied significantly according to key patient and tumor characteristics. Age analysis revealed that patients older than 65 years had a median survival time (MST) of 12.9 months (95% CI, 8.7–17.1), whereas those younger than 65 had a longer MST of 15.5 months (95% CI, 6.9–24.1), with a significant difference (*p* = 0.036) (Figure 2A). Histology-based survival showed adenocarcinoma (ADC) patients had an MST of 18.1 months (95% CI, 13.15–23.03), squamous cell carcinoma (SqC) patients 12.9 months (95% CI, 10.9–14.9), large cell carcinoma (LCC) 11.1 months (95% CI, 9.6–12.6), and those with not otherwise specified (NOS) tumors had the longest MST of 29.5 months (95% CI, 22.3–36.7), although this difference was not statistically significant (*p* = 0.230) (Figure 2B). Tumor stage analysis indicated a significant survival advantage for stage III patients, with an MST of 14.9 months (95% CI, 8.65–21.3) compared to 13.6 months (95% CI, 8.65–21.3) in stage IV patients (*p* = 0.011) (Figure 2C). Performance status by ECOG score was strongly associated with survival: patients with ECOG 0 had an MST of 22.5 months (95% CI, 12.1–32.9), ECOG 1 patients 13.3 months (95% CI, 10.4–16.3), and those with ECOG 2 had a markedly shorter MST of 3.3 months (95% CI, 2.8–3.8), with highly significant differences (*p* = 0.0001) (Figure 2D). These results highlight the impact of age, tumor stage, and functional status on survival outcomes in this population.

The multivariate Cox regression showed that, after adjusting for all other variables, only two factors were independent predictors of survival: ECOG performance status was the strongest predictor, with patients having Grade 0 or Grade 1 showing a significantly lower risk of death (HR = 0.17 and HR = 0.26, respectively; *p* < 0.01) compared to those with Grade 2; disease stage was also significant, as patients with stage III disease had a 35% lower risk of death than those with stage IV disease (HR = 0.65; *p* < 0.05). In contrast, age (whether continuous or categorical), sex, race, smoking habit, and histologic tumor type did not have a statistically significant association with survival outcomes.

The classification tree analysis identified age as the primary predictor of survival, with a cut-off of 63.5 years. The mean survival was 36.3 months for patients younger than 63.5 years and 20.6 months for older patients.

To further assess the impact of racotumomab in the real-world study, results were compared with control groups from two previous clinical trials using best supportive care or docetaxel in the maintenance scenario. Two different forest plots were created to display the results. Overall, the survival hazard ratio was 0.44 compared to supportive care (Figure 3) and 0.54 compared to docetaxel (Figure 4). Compared to supportive care, all patient subgroups except the mixed color of skin benefited from racotumomab (Figure 3). A similar pattern was observed in the comparison with docetaxel. However, variables including older age, smoking history, black skin color, and female sex showed a non-significant trend toward improved survival for racotumomab (Figure 4). Conversely, patients with ECOG performance status 2 appeared to have better survival outcomes when treated with docetaxel, although this finding did not reach statistical significance.

## 4. Discussion

Real-world evidence plays an increasing role in medical decisions, providing a valuable platform for effectiveness assessments, complementary to randomized trials [28]. This was the first large-scale real-world scenario study to assess the survival benefit of racotumomab as a switch maintenance vaccine for advanced NSCLC, after its approval by the national regulatory authority.

Carbohydrate-based antigens have been identified in various neoplasms [29]. These antigens might be specifically expressed in tumors compared to normal tissue due to the aberrant glycosylation process, as part of the malignant transformation. These gangliosides and sulfatides play a crucial role in tumor metastasis and invasiveness [30].

One strategy to overcome the low immunogenicity of carbohydrates is the use of anti-idiotypic MAbs, also known as Ab2, to function as antigen surrogates. Racotumomab is a murine IgG1 monoclonal antibody, whose idiotypic fractions mimic membrane glycoconjugates expressed in some aggressive solid tumors [16,17]. It is intended to induce an immune response against N-glycolylneuraminic acid-containing gangliosides and sulfated glycolipids [16,17]. According our previous data, racotumomab exerts an anti-angiogenic and anti-metastatic effect besides favoring the CD8^+^ T cell infiltration of the malignant tissue [18].

This retrospective, observational study evaluated racotumomab in NSCLC patients that completed the front-line therapy for the advanced stage. Treatment compliance was high, with 92.6% of patients completing induction.

The safety of racotumomab was extensively characterized in prior clinical trials involving more than 1500 patients. Consequently, given its established favorable profile, this retrospective study did not include active surveillance for adverse events. No treatment interruptions attributable to safety concerns were observed in this patient subset, and no serious adverse events (SAEs) were reported to the institutional pharmacovigilance program. These findings are consistent with the drug’s known tolerability. This is a unique characteristic of this vaccine when compared to other immunotherapies approved for NSCLC like immune checkpoint inhibitors [31,32,33,34].

The advanced NSCLC cancer patients exhibited a median survival of 14.9 months. Remarkably, although these results originate from routine clinical practice rather than controlled trials, they compare favorably with the pivotal studies that supported the vaccine’s approval and are consistent with survival outcomes reported for other continuation or switch maintenance therapies, including docetaxel (12.3 months) [35], pemetrexed (13.4 months) [36], and erlotinib (12.0 months) [37].

As anticipated, patients with less advanced disease (unresectable stage III) and better performance status (ECOG 0) exhibited superior survival, consistent with the known prognostic significance of disease burden and functional status [38,39]. The survival benefit observed in patients younger than 65 years is also expected for an active immunotherapy, which relies on the individual immune response [40,41].

Currently, anti-PD-1 antibodies, along with anti-PD-L1 antibodies, have become a cornerstone in the first-line treatment of advanced non-small cell lung cancer (NSCLC) without targetable oncogenic driver alterations [42]. This approach, works by blocking the PD-1/PD-L1 interaction, thereby reversing T-cell suppression and enabling a sustained anti-tumor immune response [43]. The integration of these agents into clinical practice has led to unprecedented prolonged survival for a significant proportion of patients with metastatic NSCLC [42,44]. Current clinical development has shifted from monotherapy in later lines to combination strategies, often with chemotherapy, in the first-line setting to improve efficacy [42,44]. For stage IV NSCLC patients without actionable mutations, median overall survival ranges from 15 to 30 months, after the use of immune-checkpoint inhibitors as monotherapy or in combination [45]. Overall survival is highly influenced by the PDL1 expression and histology [43]. Despite these advances, challenges remain in optimizing patient selection, with ongoing research focused on identifying predictive biomarkers beyond PD-L1 expression to better tailor this immunotherapeutic approach to individual patients [44]. In stage IV patients, anti-PD1 therapy is typically administered until disease progression, with overall survival measured from the start of treatment. This differs from the racotumomab study, where the vaccine was used as a switch maintenance therapy rather than from diagnosis, preventing a direct comparison of survival outcomes.

In the racotumomab data set, long-term overall survival of the treated population was very encouraging since the 5-year survival rate was 20%, irrespective of racotumomab vaccination compliance. After the use of other immunotherapies, like the anti-PD1 antibody pembrolizumab, the 5-year overall survival rate for first-line NSCLC patients (regardless of PD-L1 expression) ranges from 18–20% [31,32,33,34], which compares positively with the racotumomab data obtained in the real-world conditions.

Previously, an alternative mathematical modeling approach was applied to the racotumomab Phase II/III trial in advanced NSCLC to evaluate whether survival data were better explained by a single- or dual-population model. Both Weibull and Weibull mixture survival models demonstrated that a two-population model provided the best fit, identifying a subset of patients who experienced large benefit from the vaccine [46]. According to this analysis, the median overall survival for the long-term survivor population of racotumomab (~20% of patients) was 76.6 months [46]. The enduring effect of racotumomab was confirmed in our routine practice study.

Several attempts have been made to identify the characteristics of the long-term survivors from the anti-idiotypic vaccine. Mazorra et al. found that extended survival was linked to specific peripheral blood mononuclear cells (PBMC) biomarkers, including higher frequencies of EMRA CD8^+^ and NKT cells, elevated CD8^+^ T cell/Treg ratio, and lower baseline levels of central and effector memory CD8^+^ T cells, suggesting their potential as predictors of response [47].

These results should be interpreted with caution due to the inherent limitations of this retrospective, single-group observational study, including potential selection bias and restricted data capture. To reduce selection bias, all stage III/IV patients treated with racotumomab at the four participating research sites during the study period were included. Missing data for key demographic and tumor-related variables were minimal, so no imputations or sensitivity analyses were performed. The absence of a concurrent control group hamper establishing definitive causality or directly compare outcomes with other therapies. Another important limitation is that the standard of care for NSCLC evolved significantly since the study began; notably, none of the patients received targeted therapy or immune checkpoint inhibitors due to their unavailability in the country. Combining racotumomab with an immunomodulatory agent is a promising strategy, as clinical evidence indicates enhanced immunogenicity when anti-PD1 antibodies are administered alongside cancer vaccines [9]. Despite these limitations, the large patient cohort and extended follow-up period strengthen the external validity of our findings, consistent with the principles of real-world evidence as recognized by regulatory agencies [23,24].

## 5. Conclusions

In the era of immune checkpoint inhibitors, this study underscores the potential benefit of racotumomab, a safe active anti-idiotypic vaccine for patients with advanced NSCLC. The study confirmed that subgroups such as younger age (<65 years) and good performance status (ECOG 0–1) are associated with significantly improved survival, offering valuable guidance for clinicians in patient selection. These findings provide a strong rationale for further investigation of racotumomab in the context of modern immunotherapy. The next priorities for clinical research include new perspective, randomized, combination trials with other immunomodulatory antibodies together with the validation of the clinical and biologic predictive biomarkers.

## Figures and Tables

**Figure 1 vaccines-13-01035-f001:**
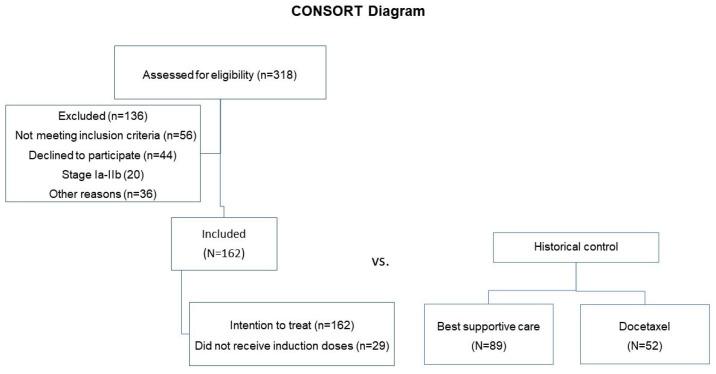
Consort diagram of patients evaluated, enrolled, treated, and included in the analysis.

**Figure 2 vaccines-13-01035-f002:**
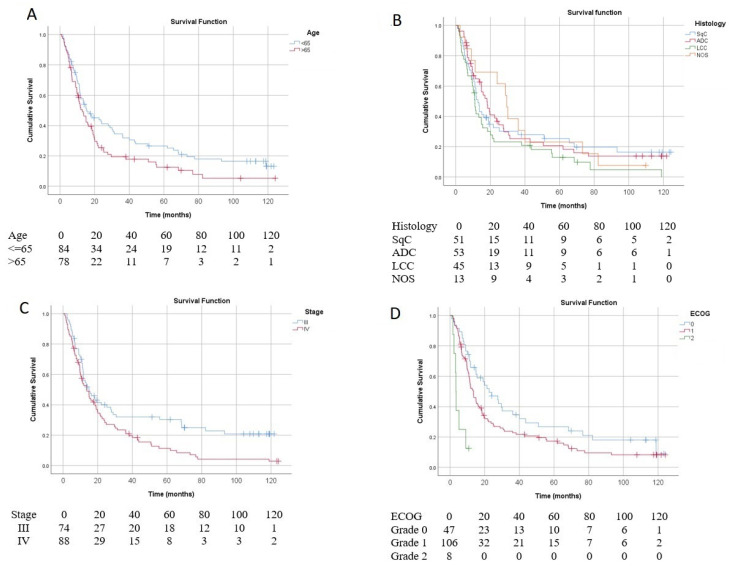
Overall survival of the ITT population according to different patient or tumor characteristics. (**A**) Survival analysis according to age. Age > 65, Median survival time (MST) = 12.9 months (95% CI 8.7–17.1); age < 65, MST = 15.5 months (95% CI 6.9–24.1) (*p* = 0.036). (**B**) Survival analysis according to histology: adenocarcinoma (ADC), MST = 18.1 months (95% CI 13.15–23.03); squamous cell carcinoma (SqC), MST = 12.9 months (95% CI 10.9–14.9); large cell carcinoma (LCC), MST = 11.1 months (95% CI 9.6–12.6); or NOS (not otherwise specified), MST = 29.5 months (95% CI 22.3–36.7) (*p* = 0.230). (**C**) Survival analysis according to tumor stage: stage III, MST = 14.9 months (95% CI 8.65–21.3) or stage IV, MST = 13.6 months (95% CI 8.65–21.3) (*p* = 0.011). (**D**) Survival analysis according to ECOG performance status: ECOG 0, MST = 22.5 months (95% CI 12.1–32.9); ECOG 1, MST = 13.3 months (95% CI 10.4–16.3); ECOG 2, MST = 3.3 months (95% CI 2.8–3.8) (*p* = 0.0001).

**Figure 3 vaccines-13-01035-f003:**
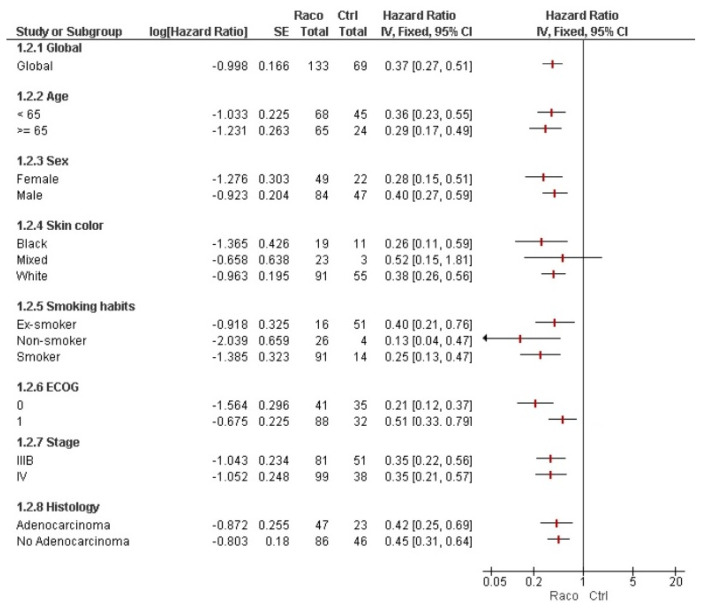
Forest plot analysis of patients treated in the real-world setting vs. control patients receiving best supportive care from a previous controlled clinical trial.

**Figure 4 vaccines-13-01035-f004:**
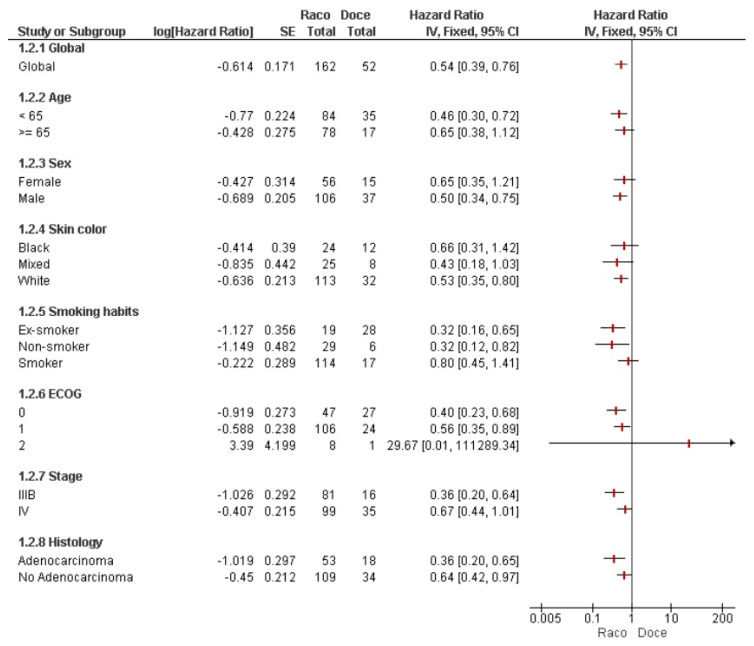
Forest plot analysis of patients treated in the real-world setting vs. docetaxel used as switch maintenance from a previous controlled clinical trial.

**Table 1 vaccines-13-01035-t001:** Patient demographics and tumor characteristics.

Demographic and Baseline Characteristics		N = 162 N (%)
Age	Mean ± SD	65 ± 10
	Median ± IQR	65 ± 14
	Minimum; Maximum	32; 89
Sex	Female	56 (34.6)
	Male	106 (65.4)
Skin color	White	113 (69.8)
	Black	24 (14.8)
	Mixed	25 (15.4)
Stage of the disease at diagnose	III (unresectable)	74 (45.7)
	IV	88 (54.3)
Histological type	Adenocarcinoma	53 (32.7)
	Large cell carcinoma	45 (27.8)
	Squamous cell carcinoma	51 (31.5)
	NOS	13 (8)
Clinical Status (ECOG)	0	47 (29.2)
	1	106 (65.8)
	2	8 (5)

Legend: ECOG: Eastern Cooperative Oncology Group; NOS: Not otherwise specified; SD: standard deviation; IQR: interquartile range.

## Data Availability

The raw data supporting the conclusions of this article will be made available by the authors upon request.

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
