# Peer review of "Real-World Effectiveness of Racotumomab as Maintenance Therapy in Advanced Non-Small Cell Lung Cancer Patients"

_vaccines, 2025, doi:10.3390/vaccines13101035_

Round 1
Reviewer 1 Report
Comments and Suggestions for Authors
This was a real-world, retrospective, observational study designed to evaluate the effectiveness of racotumomab, an anti-idiotype monoclonal antibody vaccine targeting tumor gangliosides, as maintenance therapy following first-line chemotherapy in patients with advanced NSCLC. This study is well-designed, well-organized and has reference value for similar clinical applications.
Author Response
Thank very much for the revision.
Reviewer 2 Report
Comments and Suggestions for Authors
This study provides real-world data on the use of racotumomab as maintenance therapy in advanced NSCLC, based on a large cohort and long observation period. The topic is interesting, clinically relevant and complements prior clinical trial findings. However, the following points need to be addressed:
-
The title is long, could be more concise.
-
A key limitation of the study is the retrospective observational nature. Please discuss in detail the potential sources of bias (e.g., selection of patients, missing data, lack of randomization) and how these factors may influence the interpretation of results. Please also indicate whether sensitivity analyses were performed to address incomplete datasets.
-
Adverse event reporting appears limited. Even if toxicity data were not systematically collected, any available information should be presented, and the absence of structured safety data should be acknowledged more clearly.
- The manuscript would benefit from greater detail regarding the statistical methods. How did the authors select an age cutoff of 63.5 years? Please indicate the tests used for subgroup comparisons.
-
The survival data need improvement. It would be better to put the ITT, PP, and subgroup analyses into a single comprehensive table. In addition, some figures (Kaplan–Meier curves, forest plots) need clearer labeling of axes, consistent inclusion of confidence intervals, and improved graphical quality.
-
The discussion would be strengthened by a thorough comparison with outcomes from contemporary immunotherapies, especially checkpoint inhibitors. Moreover, the identification of subgroups with longer survival is interesting. Please also discuss the translational significance of these findings, including possible biomarker approaches.
-
The conclusions should not be overstated. While the data suggest benefit from racotumomab, the limitations of retrospective design, the lack of modern comparators, and the incomplete safety profile mean that results should be interpreted cautiously. It should be put emphasis on the need for prospective trials or combinatorial strategies.
-
Please add recent reviews the References section on therapeutic cancer vaccines and immunotherapy.
-
Use consistent terminology throughout the manuscript.
-
Abbreviations (ECOG, ITT, PP) should be defined at first mention.
Author Response
The authors would like to thank the referee for the thorough review of the manuscript and for the valuable suggestions. The reviewer’s recommendations will definitely improve the article. Before a detailed reply to the referee comments, the authors would like to explain that 3 methodological changes have been incorporated into the manuscript:
- Data Set: To ensure an unbiased comparison with historical control groups, the overall survival analysis of the real-world cohort now includes only the 162 patients with stage III/IV disease. The original dataset of 182 patients included 20 patients with stage I/II (relapsing disease), who were excluded due to their more favorable prognosis. While these patients were previously excluded from the forest plot comparison, including them into the original series could have led to potential misinterpretation of survival outcomes.
- Analysis Population: All statistical analyses were conducted on an intention-to-treat (ITT) basis. Therefore, data are reported for the entire population regardless of whether patients completed the five-dose induction vaccination, preventing separate reporting for the per-protocol subset and preserving the integrity of the ITT principle.
- A new Figure, corresponding to the Consort Diagram, was added to clarify the study roadmap.
Below we reply to all the specific comments. Changes made to the manuscript were highlighted in red.
Referee 2
- The title is long, could be more concise
The title was reduced as suggested:
- A key limitation of the study is the retrospective observational nature. Please discuss in detail the potential sources of bias (e.g., selection of patients, missing data, lack of randomization) and how these factors may influence the interpretation of results. Please also indicate whether sensitivity analyses were performed to address incomplete datasets.
The following explanation was added to the discussion, as recommended.
These results should be interpreted with caution due to the inherent limitations of this retrospective, single-group observational study, including potential selection bias and restricted data capture. To reduce selection bias, all stage III/IV patients treated with racotumomab at the four participating research sites during the study period were included. Missing data for key demographic and tumor-related variables were minimal, so no imputations or sensitivity analyses were performed. The absence of a concurrent control group limits our ability to establish definitive causality or directly compare outcomes with other therapies. Another important limitation is that the standard of care for NSCLC evolved significantly since the study began; notably, none of the patients received targeted therapy or immune checkpoint inhibitors due to their unavailability in the country
- Adverse event reporting appears limited. Even if toxicity data were not systematically collected, any available information should be presented, and the absence of structured safety data should be acknowledged more clearly.
The following explanation was added to material and methods:
Since the safety of this drug had already been thoroughly evaluated in previous clinical trials, the collection of data on adverse events was not an objective of this observational, retrospective study. After the marketing approval, given its favorable safety profile, a passive pharmacovigilance system was implemented, relying on spontaneous reports of serious adverse events.
and to the discussion:
The safety of racotumomab had been extensively characterized in prior clinical trials involving more than 1,500 patients. Consequently, given its established favorable profile, this retrospective study did not include active surveillance for adverse events. No treatment interruptions attributable to safety concerns were observed in this patient subset, and no serious adverse events (SAEs) were reported to the institutional pharmacovigilance program. These findings are consistent with the drug's known tolerability. This is a unique characteristic of this vaccine when compared to other immunotherapies approved for NSCLC like immune checkpoint inhibitors.
- The manuscript would benefit from greater detail regarding the statistical methods. How did the authors select an age cutoff of 63.5 years? Please indicate the tests used for subgroup comparisons.
These elements were added to Material and methods and a new reference on the classification and regression tree analysis was included:
Furthermore, a classification and regression tree (CART) analysis was employed to identify subgroups of patients with better survival outcomes based on all baseline variables. This non-parametric method uses a process of recursive binary partitioning to hierarchically split the study population into increasingly homogeneous subgroups. The algorithm selects the cutoff that creates the most distinct groups based on the survival outcome (27). The CART model employed a growing method with overall survival (continuous) as the dependent variable. Independent variables included age (continuous), sex, skin color, smoking habit, ECOG performance status, histology, and stage, all treated as categorical variables. The maximum tree depth was set to 5 nodes.
Regarding the tests used for subgroup comparisons, as previously explained in the manuscript, survival curves were compared using the log-rank test with a significance level of α=0.05.
- The survival data need improvement. It would be better to put the ITT, PP, and subgroup analyses into a single comprehensive table. In addition, some figures (Kaplan–Meier curves, forest plots) need clearer labeling of axes, consistent inclusion of confidence intervals, and improved graphical qualit
As explained, the analysis was conducted on an intention-to-treat basis only; and the per-protocol analysis was not reported. The baseline characteristics of this population are summarized in Table 1. Furthermore, the survival figures were improved and modified to display the number of patients at risk over time. The corresponding legends were modified to clearly indicate the median survival, including 95% confidence intervals, for each study group, plus the p value for each comparison.
- The discussion would be strengthened by a thorough comparison with outcomes from contemporary immunotherapies, especially checkpoint inhibitors. Moreover, the identification of subgroups with longer survival is interesting. Please also discuss the translational significance of these findings, including possible biomarker approaches.
New elements on the effect of checkpoints inhibitors were added, as recommended.
Currently, anti-PD-1 antibodies, along with anti-PD-L1 antibodies, have become a cornerstone in the first-line treatment of advanced non-small cell lung cancer (NSCLC) without targetable oncogenic driver alterations (42). This approach, works by blocking the PD-1/PD-L1 interaction, thereby reversing T-cell suppression and enabling a sustained anti-tumor immune response (43). The integration of these agents into clinical practice has led to unprecedented prolonged survival for a significant proportion of patients with metastatic NSCLC (42, 44). Current clinical development has shifted from monotherapy in later lines to combination strategies, often with chemotherapy, in the first-line setting to improve efficacy (42, 44). For stage IV NSCLC patients without actionable mutations, median overall survival ranges from 15 to 30 months, after the use of immune-checkpoints inhibitors as monotherapy or in combination (45). Overall survival is highly influenced by the PDL1 expression and histology (43). Despite these advances, challenges remain in optimizing patient selection, with ongoing research focused on identifying predictive biomarkers beyond PD-L1 expression to better tailor this immunotherapeutic approach to individual patients (44). In stage IV patients, anti-PD1 therapy is typically administered until disease progression, with overall survival measured from the start of treatment. This differs from the racotumomab study, where the vaccine was used as a switch maintenance therapy rather than from diagnosis, preventing a direct comparison of survival outcomes.
Regarding the racotumomab predictive biomarkers, it was previously stated that patients that survived longer had distinct immune cell profiles, suggesting these could be used to predict treatment response. Next steps involve testing combination therapies with other immunomodulatory antibodies and validating predictive biomarkers.
- The conclusions should not be overstated. While the data suggest benefit from racotumomab, the limitations of retrospective design, the lack of modern comparators, and the incomplete safety profile mean that results should be interpreted cautiously. It should be put emphasis on the need for prospective trials or combinatorial strategies.
It was further highlighted that these results should be interpreted with caution, considering the inherent limitations of the real-world study and that executing new perspective, randomized, combination trials with other immunomodulatory antibodies together with the validation of the clinical and biologic predictive biomarkers is the next priority.
The abstract was also modified to temper the results.
- Please add recent reviews the References section on therapeutic cancer vaccines and immunotherapy.
New elements on cancer vaccines and other immunotherapies were added, as suggested. New references were inserted accordingly.
- Use consistent terminology throughout the manuscript.
The terminology was carefully reviewed and corrected throughout the entire manuscript.
- Abbreviations (ECOG, ITT, PP) should be defined at first mention.
Abbreviations were revised in order to grant defining them at first mention.
Reviewer 3 Report
Comments and Suggestions for Authors
The paper describes the effects of Racotumomab, an antibody inducing in a patient, when immunized with adjuvant, antibodies to ganglioside GM3, the latter expressed by tumor cells. The authors selected two groups of treatment: the patients recieved different numbers of immunizations called intention to treatment (ITT) with average 13 immunizations and per protocol PP group with average 18 injections. The results are mostly shown for ITT group and not compared with PP one (or not clearly described). Figure 1 must be improved, fonts increased, legends corrected. See the pdf file for other remarks.

Author Response
The authors would like to thank the referee for the thorough review of the manuscript and for the valuable suggestions. The reviewer’s recommendations will definitely improve the article. Before a detailed reply to the referee comments, the authors would like to explain that 3 methodological changes have been incorporated into the manuscript:
- Data Set: To ensure an unbiased comparison with historical control groups, the overall survival analysis of the real-world cohort now includes only the 162 patients with stage III/IV disease. The original dataset of 182 patients included 20 patients with stage I/II (relapsing disease), who were excluded due to their more favorable prognosis. While these patients were previously excluded from the forest plot comparison, including them into the original series could have led to potential misinterpretation of survival outcomes.
- Analysis Population: All statistical analyses were conducted on an intention-to-treat (ITT) basis. Therefore, data are reported for the entire population regardless of whether patients completed the five-dose induction vaccination, preventing separate reporting for the per-protocol subset and preserving the integrity of the ITT principle.
- A new Figure, corresponding to the Consort Diagram, was added to clarify the study roadmap.
Below we reply to all the specific comments. Changes made to the manuscript were highlighted in red.
Referee 3
- The results are mostly shown for ITT group and not compared with PP one (or not clearly described). Figure 1 must be improved, fonts increased, legends corrected. See the pdf file for other remarks.
The per protocol analysis was deleted to avoid unintended bias. Former Figure 1 (current Figure 2) was corrected and the legend was expanded to include all median survival times and p-values.
Former Figure 1 B (now 2B) changed with the new data set. However, no significant survival differences were found among the different histologies (p=0,230). Pairwise comparisons between individual curves should not be done, if the overall log-rank test p-value is greater than 0.05. Respecting the non-significant overall test is a fundamental principle of good statistical practice that prevents overinterpreting chance findings.
All the remaining suggestions in the manuscript were implemented.
Reviewer 4 Report
Comments and Suggestions for Authors
This manuscript addresses an important question: how racotumomab performs outside clinical trials in a real-world setting where modern targeted and immune therapies were largely unavailable. This retrospective multicenter study reports outcomes for 182 advanced NSCLC patients treated with racotumomab across four Cuban hospitals. The cohort is sizeable, and the survival outcomes are potentially meaningful. This study could make a useful contribution to literature, particularly for settings with limited access to checkpoint inhibitors. The dataset is potentially valuable, but the manuscript currently has several core methodological, analytical, and reporting shortcomings that could affect interpretation. The main concerns are immortal-time bias, lack of adjusted survival modelling, incomplete handling of missing data, and unclear methods for comparisons with historical controls.
Here are my detailed comments and suggestions:
- Present multivariable Cox PH models (adjusted HRs) with 95% CIs and p-values. List covariates used, rationale for inclusion, and modelling strategy.
- Show proportional hazards test results or present alternative modelling
- Report median follow-up, and provide Kaplan-Meier plots with numbers at risk and censoring marks.
- For the PP population, include a landmark analysis or modelling with induction completion as time-dependent to avoid immortal time bias.
- Provide details of classification tree methods and performance metrics
- Describe precisely which historical trials / control arms were used and extraction method
- Add sensitivity analyses that restrict the cohort to patients with baseline characteristics closest to trial populations.
- Add a CONSORT-style flow diagram detailing numbers screened/enrolled/excluded and reasons.
- Add a table summarizing the extent of missing data per variable and handling strategy.
- Describe pharmacovigilance procedures; provide AE/SAE table using CTCAE grades if possible.
- Temper conclusions: avoid definitive claims of superiority to modern checkpoint inhibitors without head-to-head comparisons.
- Expand the limitations section to clearly outline biases inherent to retrospective real-world studies.
Author Response
The authors would like to thank the referee for the thorough review of the manuscript and for the valuable suggestions. The reviewer’s recommendations will definitely improve the article. Before a detailed reply to the referee comments, the authors would like to explain that 3 methodological changes have been incorporated into the manuscript:
- Data Set: To ensure an unbiased comparison with historical control groups, the overall survival analysis of the real-world cohort now includes only the 162 patients with stage III/IV disease. The original dataset of 182 patients included 20 patients with stage I/II (relapsing disease), who were excluded due to their more favorable prognosis. While these patients were previously excluded from the forest plot comparison, including them into the original series could have led to potential misinterpretation of survival outcomes.
- Analysis Population: All statistical analyses were conducted on an intention-to-treat (ITT) basis. Therefore, data are reported for the entire population regardless of whether patients completed the five-dose induction vaccination, preventing separate reporting for the per-protocol subset and preserving the integrity of the ITT principle.
- A new Figure, corresponding to the Consort Diagram, was added to clarify the study roadmap.
Below we reply to all the specific comments. Changes made to the manuscript were highlighted in red.
- Present multivariable Cox PH models (adjusted HRs) with 95% CIs and p-values. List covariates used, rationale for inclusion, and modelling strategy.
The multivariate Cox regression analysis was added to the manuscript (see material and method & Results), as follows:
Material and methods:
A multivariate Cox proportional hazards regression model was used to assess the independent impact of various clinical and demographic factors on patient survival time. The analysis was conducted by entering all variables, including age (analyzed both as a continuous measure and a binary categorical variable), sex, skin color, smoking habit, ECOG performance status, histologic type, and disease stage simultaneously into the model.
Results
The multivariate Cox regression revealed that, after adjusting for all other variables, only two factors were independent predictors of survival: ECOG performance status was the strongest predictor, with patients having Grade 0 or Grade 1 showing a significantly lower risk of death (HR = 0.17 and HR = 0.26, respectively; p < 0.01) compared to those with Grade 2; disease stage was also significant, as patients with stage III disease had a 35% lower risk of death than those with stage IV disease (HR = 0.65; p < 0.05). In contrast, age (whether continuous or categorical), sex, race, smoking habit, and histologic tumor type did not have a statistically significant association with survival outcomes.
- Show proportional hazards test results or present alternative modelling
Evaluation of proportional hazards tests.
For the comparison vs. best supportive care, all subgroups but one showed no violation (p-values > 0.05) of the proportional hazard assumption. However, for the ECOG=2 subgroup, the ph_test p-value is 0.009, which was highly significant. This particular subgroup was deleted from the forest plot.
For the comparison vs. docetaxel, the hazard ratios (HRs) for all subgroups showed p-values > 0.05 and had a consistent effect over time.
- Report median follow-up, and provide Kaplan-Meier plots with numbers at risk and censoring marks.
It was clarified that the median follow-up time of the patients was 112 months. The Kaplan-Meier plots were completed with numbers at risk and censoring marks.
- For the PP population, include a landmark analysis or modelling with induction completion as time-dependent to avoid immortal time bias.
In the current manuscript version, statistical analyses were conducted only on an intention-to-treat (ITT) basis. The PP analysis was omitted from the manuscript to avoid immortal time bias. In the ITT analysis, as described, all patients were followed from racotumomab first vaccination.
- Provide details of classification tree methods and performance metrics
The classification tree methods and performance metrics details were provided, as suggested.
- Describe precisely which historical trials / control arms were used and extraction method.
To assess the impact of racotumomab in the real-world setting across different subgroups, the study database was integrated with data from the control arms of prior controlled racotumomab trials. The historical data used for contextual comparison were derived from two pivotal, controlled, clinical trials of racotumomab: Phase II/III trial (ID: RPCEC00000009; https://rpcec.sld.cu/trials/RPCEC00000009-En) (21) and the Phase III study (ID: RPCEC000000179; https://rpcec.sld.cu/trials/RPCEC00000179-En) (23). Data extraction was accurately done, with overall survival, patient demographics, and baseline characteristics obtained directly from the original databases. The process was performed by the same statistician involved in the original studies, ensuring consistent and direct oversight.
- Add sensitivity analyses that restrict the cohort to patients with baseline characteristics closest to trial populations.
To ensure an unbiased comparison with historical control groups, the current manuscript of the real-world cohort now includes only the 162 patients with stage III/IV disease, which are strictly comparable to the previous trial populations. Consequently, according the new manuscript version, no patients were excluded and sensitivity analysis is not needed.
- Add a CONSORT-style flow diagram detailing numbers screened/enrolled/excluded and reasons.
A CONSORT-style flow diagram was added, as suggested.
- Add a table summarizing the extent of missing data per variable and handling strategy.
Missing data for key demographic and tumor-related variables were minimal, so no imputations or sensitivity analyses were performed. Table 1 describes patient demographics and tumor characteristics.
- Describe pharmacovigilance procedures; provide AE/SAE table using CTCAE grades if possible.
The following explanation was added to material and methods:
Since the safety of racotumomab was thoroughly evaluated in previous large clinical trials, involving more than 1500 NSCLC patients, the collection of data on adverse events was not an objective of this observational, retrospective study. After the marketing approval, given racotumomab favorable safety profile, a passive pharmacovigilance system was implemented, relying on spontaneous reports of serious adverse events.
and to the discussion:
The safety of racotumomab had been extensively characterized in prior clinical trials involving more than 1,500 patients. Consequently, given its established favorable profile, this retrospective study did not include active surveillance for adverse events. No treatment interruptions attributable to safety concerns were observed in this patient subset, and no serious adverse events (SAEs) were reported to the institutional pharmacovigilance program. These findings are consistent with the drug's known tolerability. This is a unique characteristic of this vaccine when compared to other immunotherapies approved for NSCLC like immune checkpoint inhibitors.
- Temper conclusions: avoid definitive claims of superiority to modern checkpoint inhibitors without head-to-head comparisons.
The conclusions were tempered in the abstract and in the discussion.
- Expand the limitations section to clearly outline biases inherent to retrospective real-world studies.
The section on the study limitation were further expanded, as suggested.
Round 2
Reviewer 2 Report
Comments and Suggestions for Authors
Thank you for answering all comments.
Author Response
Thanks a lot for the deep revision!!
Reviewer 3 Report
Comments and Suggestions for Authors
The authors removed two observation groups which is good as before the groups were not clearly presented, added some explanation. My major remark was to the figure 2 which still needs correction: please remove streak mark "censurado" from figure body and add that comment to the figure legend. I guess that these streaks corresponds to the injections of antibody while nowhere this is explained.

Author Response
Thanks again for the review. Figure 2 was corrected as suggested. Anyway, the authors would like to clarify that “censored” refers to subjects whose individual survival time cannot be accurately determined because the event of interest (death) have not occur at the time of the survival analysis. Essentially, these patients were alive at the time of the analysis. Censoring marks (vertical lines in the survival surves) are mandatory in any Kaplan-Meier curve and indicate how long the surviving subjects were followed up.